# FedBridge: Bridging Domain Experts and Domain Knowledge via a Federated Learning Framework for Controlled Model Personalization

## Abstract

The rapid development of large language models (LLMs) has highlighted a critical challenge in applying these models to domain-specific tasks while preserving data privacy. This study introduces FedBridge, a novel architecture that seamlessly bridges parameter-efficient fine-tuning (PEFT) and retrieval-augmented generation (RAG) within a federated learning framework, enabling the deep integration of domain expert and knowledge through these three pillars. Initially, we propose FF-LoRA (Federated Fusion Low-Rank Adaptation), a PEFT variant that fuses server-level global representations with client-specific parameters to mitigate client drift caused by heterogeneous local data. Following this, we design a dual-task strategy that constructs independent local case and global authoritative bases, enabling independent querying and targeted retriever optimization. Furthermore, we establish bidirectional consistency between the fine-tuned domain models and the retriever system: the domain model's output guides retriever to precisely identify latent relevant documents, while concurrently serving as a generator, thus improving coherence and domain fidelity of the retrieval-response pipeline. Experimental results demonstrate that the proposed architecture efficiently improves accuracy and robustness in both close-ended domain and open-ended domain tasks.

## 1 Introduction

Despite the success of large language models (LLMs) in a range of natural language processing tasks, they face substantial challenges when applied to domain-specific applications. These challenges are particularly acute as publicly available datasets continue to shrink Villalobos et al. (2022), while domain-specific data, due to privacy concerns, remains decentralized and fragmented. This issue significantly limits the application of general LLMs Touvron et al. (2023a;b) in industries such as healthcare, where data aggregation is not feasible due to legal privacy restrictions. For instance, in the healthcare domain Han et al. (2023); Zhang et al. (2023), medical data from disparate institutions is often fragmented and siloed, preventing effective model training. While Federated Learning (FL) frameworks McMahan et al. (2017) address privacy concerns by decentralizing data, they exacerbate problems related to data heterogeneity, particularly when substantial variations in data types and volumes exist across clients Li et al. (2019). As illustrated in Figure 1(a), the publicly available HealthCareMagic-100K dataset Li et al. (2023) exhibits significant variation in data distribution across categories. Real-world data distributions are even more complex, as hospitals differ in scope and specialization, further exacerbating data distribution heterogeneity. Figure 1(b) illustrates the simulated data distribution across 10 hospitals based on Figure 1(a), including both general and specialized institutions.

Although FL framework can protect privacy and optimize the model training, as shown in Figures 1(b) to 1(a), aggregating data from multiple hospitals to train a powerful global model that encompasses a wide range of domain knowledge, the global model produces often underperform on domain-specific tasks, especially when these tasks constitute a small portion of the overall data

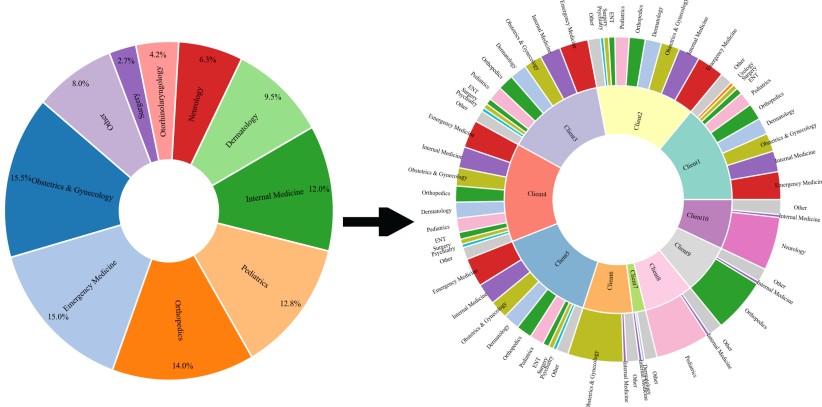

Figure 1: Data distribution in the medical domain dataset HealthCareMagic-100K. (a) Distribution of questions and answers across 10 categories. (b)Simulated data distribution across 5 general hospitals and 5 specialized hospitals.

Zhao et al. (2018). This challenge underscores the necessity of personalized fine-tuning, which adapts models to specific tasks and user requirements.

Furthermore, while full fine-tuning enables rapid adaptation to domain-specific tasks, its computational demands often exceed the capabilities of federated clients. In contrast, parameter-efficient fine-tuning (PEFT) involves training only a small number of parameters while keeping the general knowledge of the foundational model frozen Hu et al. (2022). This efficiency makes PEFT an attractive solution for federated learning settings. However, PEFT methods face limitations by excessive parameter sparsity, which impedes the model's ability to effectively capture domain-specific features, ultimately affecting model performance Fu et al. (2023); Li et al. (2025). Notably, FlexLoRA Bai et al. (2024) demonstrates that high-rank better captures general knowledge, while low-rank excels at task-specific feature extraction.

Another prevalent issue in domain-specific LLMs is the "hallucination" phenomenon, in which models generate unreliable or irrelevant outputs. Retrieval-augmented generation (RAG) mitigates this by incorporating external knowledge sources. For instance, FRAG Zhao (2024) significantly enhances the model's accuracy in complex tasks. However, RAG-based LLMs still encounter challenges, including: (1) general LLMs acting as generators without a deep understanding of domain-specific documents, (2) vague user queries leading to suboptimal retrieval, (3) inconsistencies between retriever and generator components, diminishing the overall quality of responses.

To address the aforementioned challenges encountered in domain-specific application, we propose the FedBridge, which has three pillars of advantage. We delineate the primary contributions of our research as follows:

- We present FF-LoRA, which integrates personalized client features with global server-side information, alleviating client drift in FL framework.

- We implement the fusion training of domain expert and knowledge provided by PEFT and RAG, respectively, within the FL framework. This approach enables fine-grained alignment between retriever and generator components, with the Domain-LLM acting both as a generator and as a guide for retriever.

- By constructing domain-specific case and authoritative knowledge bases, we implement a dual-task strategy that optimizes the controlled and efficient application of LLMs for domain-specific tasks. Experimental results validate the effectiveness of FedBridge in improving performance across both general and domain-specific evaluation metrics.

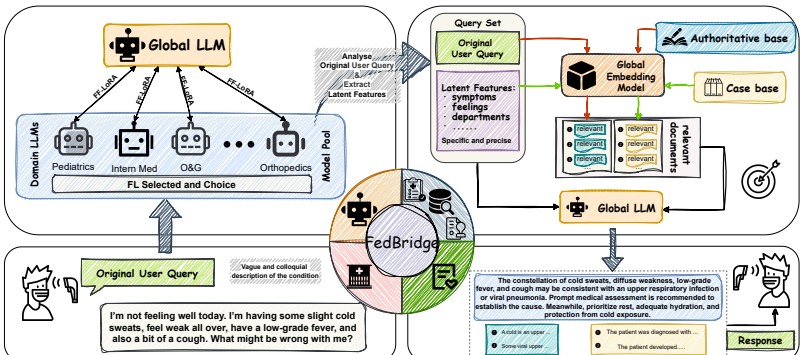

Figure 2: Overall FedBridge architecture in the medical domain. Upon submission of a query, an appropriate domain-specific model is selected from the FF-LoRA trained federated model pool. Latent features of the query are extracted through instruction mining and subsequently sent to the retriever for embedding processing, which retrieves the top-K relevant documents. These documents are then processed by the selected domain-specific model.

## 2 MOTIVATION

Existing PEFT methods face limitations due to excessive sparsity, which hinders their ability to effectively capture domain-specific knowledge and diminishes the model's generalization capabilities. Moreover, domain models trained through FL and PEFT are often affected by client drift. While FL facilitates data aggregation across multiple clients to form a powerful global model, inconsistencies between local client tasks and the global model persist, undermining performance on domain-specific tasks.

To address these challenges, our research aims to integrate federated global features with client-specific personalized features. By leveraging distinct models for various tasks within the FL model pool, we aim to minimize the negative effects of data distribution heterogeneity on model performance. Furthermore, by expanding the training parameters, we seek to alleviate the sharpness issues associated with PEFT sparsity techniques, thereby enhancing the model's generalization capacity.

More critically, despite the advancements brought about by PEFT, domain-specific models still encounter the "hallucination" problem when handling domain-specific queries. Current RAG systems are similarly plagued by the "garbage in, garbage out" issue, where colloquial or vague user queries lead to poor retrieval quality and, consequently, low-quality generated outputs. Additionally, inconsistencies between the retriever and generator exacerbate difficulties in accurately interpreting specialized documents, further degrading the quality of the generated content.

In response to these issues, we propose the FedBridge architecture, which builds upon the foundations of FL and PEFT. Our approach incorporates a RAG-based framework that ensures a precise mapping between user queries, retrieved documents, and generated responses. Collaborative training ensures robust knowledge consistency between the retriever and generator components

## 3 METHODOLOGY

We propose the FedBridge framework, a novel architecture that integrates FL, PEFT, and RAG, as illustrated in Figure 2. To address client drift resulting from data heterogeneity, we introduce FF-LoRA, a method designed to fuse global features from FL aggregation with personalized client features. This approach creates a federated model pool that retains personalized characteristics of each client while maintaining global coherence.

In domain-specific applications, using PEFT or RAG independently often fails to address task complexity effectively. Therefore, we combine the strengths of both techniques to leverage external non-parametric knowledge alongside internal parametric knowledge. This fusion strategy alleviates the challenges faced by LLMs in complex application scenarios. Specifically, for knowledge-intensive

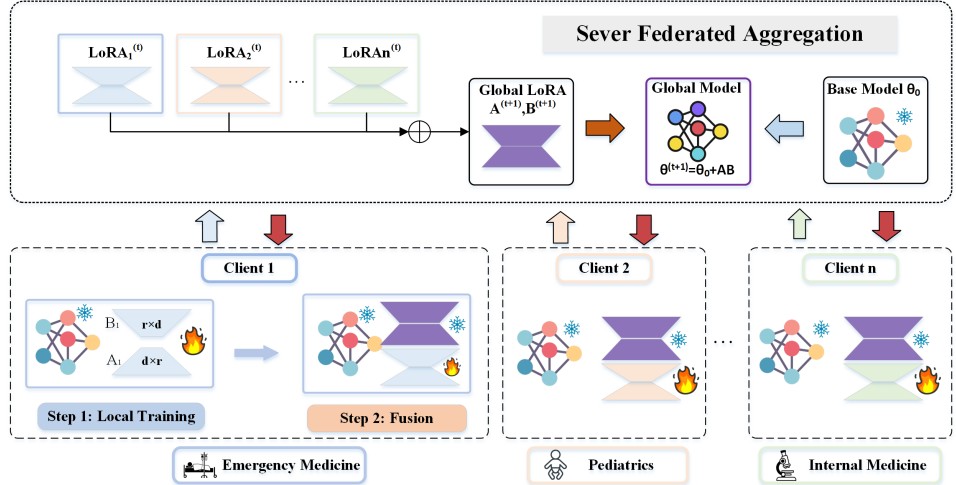

Figure 3: FF-LoRA integration of global and personalized client features to mitigate client drift in FL framework.

tasks, we provide information through an authoritative and case base to build "Knowledge Boundary " to limit the LLM hallucination problem. Meanwhile, the domain-specific LLM functions as both the comprehension module for the RAG retrieval and as the final generator, ensuring consistent preferences between the retrieval and the generation based on a shared understanding of domain-specific knowledge.

## 3.1 FEDERATED TWO-STAGE KNOWLEDGE FUSION STRATEGY

To address the client drift induced by data heterogeneity in existing FL and PEFT methods, as well as the sharpness problem arising from the limited parameters in LoRA Hu et al. (2022), we propose the FF-LoRA, which incorporates a two-stage knowledge fusion strategy. Initially, FF-LoRA integrates the global features understood by the server model with the personalized client features enabling the model to perform specific tasks based on global understanding. This fusion enhances the model's ability to grasp domain knowledge while focusing on task-specific aspects.

The architecture of our proposed algorithm is depicted in Figure 3, the FF-LoRA consists of two stages. In the first stage, each client defines its unique $A$ and $B$ matrices, initialized similarly to LoRA. Then, parameters from all clients are globally aggregated, represented as $A = \{A_1, A_2, \ldots, A_n\}$ and $B = \{B_1, B_2, \ldots, B_n\}$. After local training on client datasets, each client uploads its $A$ and $B$ matrices to the global server for aggregation. The server aggregates these matrices to form the global model, denoted as $Global\_C$ and $Global\_D$, which represents the fused knowledge from all clients and is sent back to each client.

In the second stage, each client fuses its local model with the global model. During this fusion, the global LoRA model is frozen, similar to pre-trained weights, preserving global knowledge and reducing computational overhead during training. Notably, this method builds on the findings of Bai et al. (2024), where high-rank LoRA captures general knowledge more effectively, while low-rank LoRA excels in task-specific feature extraction.

To achieve this functionality, we use the formula to explain it in detail. During stage 1 training, the training process of each client is formulated as follows,

$$h = W_0 x + \Delta W x = W_0 x + BA x \tag{1}$$

where $W_0 \in \mathbb{R}^{d \times d}$ represents the frozen pre-trained weights, $\Delta W$ represents a minimal training matrix set, and $A \in \mathbb{R}^{r \times d}$ and $B \in \mathbb{R}^{d \times r}$ are low-rank matrices. The $r$ represents the rank, and $r \ll d$, thereby reducing the resource requirements. After training, each client uploads its weights to the global server.

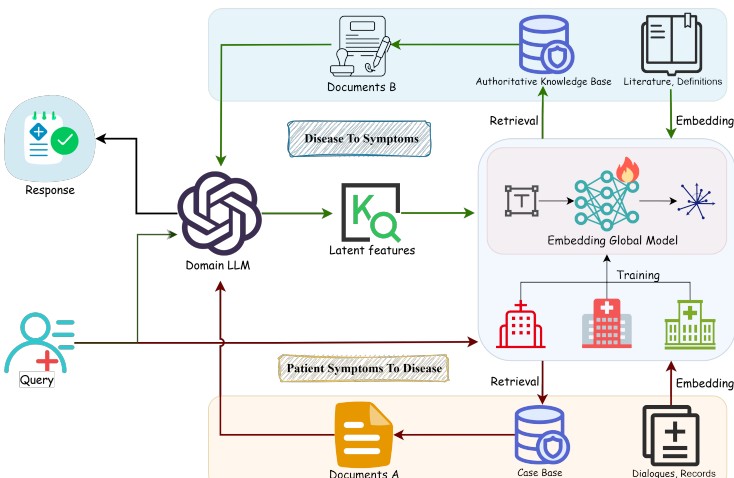

Figure 4: FedBridge integrates FF-LoRA with RAG for enhancing domain-specific tasks with external knowledge.

In stage 1, the global server aggregates the low-rank matrices from all clients. The server aggregation is formulated as follows,

$$\sum_{n=0}^{N} Global\_C_n Global\_D_n = (B_0 \oplus \cdots \oplus B_N)(A_0 \oplus \cdots \oplus A_N) \tag{2}$$

where $Global\_C \in \mathbb{R}^{c \times d}$ and $Global\_D \in \mathbb{R}^{d \times c}$ represent the global model formed by aggregating the $A$ and $B$ matrices from all clients. The aggregated high-rank $c$ capture the global general features. After the first round of collaborative training, the global server sends the aggregated global model weight back to each client.

In stage 2, each client fuses its local weights $A$ and $B$, with the global weights $Global\_C$ and $Global\_D$. The fusion is performed as follows,

$$h = W_0 x + \Delta W = W_0 x + \Delta_g W \Delta_c W x \tag{3}$$

where $\Delta_g W$ and $\Delta_c W$ are the weights from the client and server, respectively, which update the original training parameter $\Delta W$. Once the weights are fused, the framework proceeds with Stage 2 training, which is expressed as follows,

$$\Delta W = \Delta_g W \Delta_c W x = F_{\text{global}} F_{\text{personalized}} x = Global\_D Global\_C B A x \tag{4}$$

where $F_{\text{global}}$ and $F_{\text{personalized}}$ represent the global features and personalized client features, respectively. After client training, only the low-rank matrices $A$ and $B$ are uploaded, and the global server aggregates and updates the matrices. Through two rounds of training, the model achieves global understanding while retaining personalized features, without requiring reinitialization of client weights later.

By utilizing FF-LoRA, we expand the parameters of LoRA, alleviating the sharpness issue inherent in the original method. The two-stage training strategy also mitigates instability caused by frequent initialization, improving both training efficiency and model stability.

## 3.2 KNOWLEDGE BOUNDARY ESTABLISHMENT

In medical applications, diagnostic accuracy is critical, yet current LLMs often suffer from hallucination problem. To address this, we introduce the FedBridge architecture, integrating FF-LoRA with RAG. This architecture systematically incorporates fragmented medical data—such as case records, manuals, dialogues, and literature—into a cohesive knowledge base. The RAG-formed knowledge boundaries act as a safeguard against hallucinations in LLMs. Furthermore, based on FL framework deeply integrates domain expert and knowledge, establishing a new paradigm for intelligent healthcare.

As illustrated in Figure 4, we construct two distinct knowledge bases: the Authoritative Knowledge Base, which contains authoritative disease definitions, treatment protocols, and literature, and the Case Knowledge Base, which includes doctor-patient dialogues and structured case reports. Each hospital (client) maintains its own case base, differing in data quantity and type, while all hospitals share the authoritative base, representing public medical standards.

To mitigate the hallucination issue in LLMs and leverage the diagnostic reasoning process of clinical doctors, we propose an understanding-based inference strategy within the FedBridge. This strategy improves model ability through dual-stage knowledge acquisition and enhancement. The domain-specific enables the model to better understand the latent domain knowledge in user queries, while the rule-based retrieval from the knowledge base helps mitigate hallucinations.

For accurate query interpretation, we extract latent features from user queries using the domain-specific LLM. The formula is as follows,

$$Latent = \mathrm{f}_{\mathrm{domain}}(q; IT) \tag{5}$$

where, $\mathrm{f}_{\mathrm{domain}}$ represents the domain-specific LLM, processed by FF-LoRA to process the query $q$ in an instruction template $IT$, and the output is the latent features $Latent$. These features, along with the original query, form two parallel query conditions: one for retrieving from the authoritative knowledge base and the other for retrieving from the case base. The formulas are as follows,

$$\begin{aligned}
\mathbf{e}_q &= E(q; \Theta), \ \mathbf{e}_L = E(Latent; \Theta), \\
D_{\mathrm{auth}} &= \mathrm{TopK}(\mathbf{e}_L, \mathcal{C}_{\mathrm{auth}}), \\
D_{\mathrm{case}} &= \mathrm{TopK}(\mathbf{e}_q, \mathcal{C}_{\mathrm{case}}^i), \quad i = 1, \ldots, M.
\end{aligned} \tag{6}$$

where $E(\cdot; \Theta)$ is the globally shared retriever embedding function applied to both the original query and latent features. Each client fine-tunes the embedding module locally based on its own case data and aggregates the updated model to the global server. $\mathcal{C}_{\mathrm{auth}}$ and $\mathcal{C}_{\mathrm{case}}^i$ represent the authoritative knowledge base and the case base of client $i$, respectively.

Finally, the answer is generated by the domain-specific LLM as the generator $G$, based on the retrieved results, the final corresponding result is formulated as follows,

$$y = \mathrm{G}(D_{\mathrm{auth}}, D_{\mathrm{case}}, q) \tag{7}$$

where $D_{\mathrm{auth}}$ and $D_{\mathrm{case}}$ represent the relevant document from dual knowledge bases. Notably, the authoritative knowledge base is used to define the characteristics of diseases (e.g., the characteristic of a disease are specific "symptoms"), while the hospital case base stores doctor-patient dialogues and medical records (e.g., the symptoms presented by the patient lead to the diagnosis of a particular disease). These two types of knowledge bases complement each other through a bidirectional mapping, specifically between disease characteristics and symptoms, as well as between symptoms and diagnostic outcomes, thereby enhancing the model's deep understanding and reasoning ability regarding disease features.

To address potential inconsistencies between the retriever and generator preferences during RAG processes, we jointly train them using the same dataset, while helps bridge the gap between general scenarios and medical-specific needs. It also ensuring that retrieved documents and generated answers are aligned for greater consistency in medical contexts.

## 4 EXPERIMENTS

To evaluate the effectiveness and robustness of our approach, we conducted extensive comparisons with state-of-the-art methods across multiple datasets. The experiments were run on eight A100 (40GB) GPUs, and the code has been publicly released at https://anonymous.4open.science/r/F-63AD.

### 4.1 EXPERIMENTAL SETUP

Initially, we evaluated the performance across a range of scenarios, considering both general and medical applications. In the general scenario, we compared FF-LoRA with state-of-the-art methods

Table 1: Performance Comparison of FF-LoRA and Baseline Methods on MMLU and MT-Bench

| Model | Policy | Method | MMLU | | | MT-Bench | |
|---|---|---|---|---|---|---|---|
| | | | Dolly | Alpaca | Wizard | Wizard | ShareGPT |
| Tiny LLama | Centralized | LoRA | 28.34 | 27.86 | 30.07 | 2.37 | 2.72 |
| | Home | FedIT | 16.93 | 30.27 | 42.34 | 3.07 | 2.53 |
| | | FFLoRA | 32.76 | 33.11 | 45.13 | 3.11 | 2.7 |
| | Heter | FLoRA | 18.43 | 29.71 | 41.43 | 3.34 | 2.64 |
| | | FFLoRA | 19.69 | 31.68 | 44.27 | 3.4 | 2.97 |
| Llama | Centralized | LoRA | 36.91 | 28.76 | 31.55 | 4.31 | 4.03 |
| | Home | FedIT | 29.61 | 29.45 | 33.47 | 3.27 | 3.75 |
| | | FFLoRA | 31.75 | 30.38 | 35.1 | 4.02 | 3.98 |
| | Heter | FLoRA | 28.62 | 29.73 | 27.7 | 4.16 | 3.59 |
| | | FFLoRA | 30.4 | 31.51 | 29.55 | 4.42 | 3.81 |

on several public datasets, encompassing diverse experimental settings (homogeneous, heterogeneous, multi-node, and single-node). This enabled a comprehensive evaluation of the proposed FF-LoRA base operator in varied environments. The training datasets included Dolly Zhang et al. (2024a), Alpaca Taori et al. (2023), Wizard Luo et al. (2023), and ShareGPT, while the evaluation datasets included MMLU Hendrycks et al. (2020) and MT-Bench Zheng et al. (2023).

In addition, we applied the FedBridge architecture to medical applications, where federated learning is crucial for data privacy. We compared FedBridge with existing domain LLMs in the healthcare domain and other federated learning-based models. The medical evaluation datasets included MedQA Jin et al. (2021), PubMedQA Jin et al. (2019), MedMCQA Pal et al. (2022) (closed-ended Q&A), and iCliniq (open-ended Q&A), while the training dataset consisted of HealthCareMagic-100K. The baseline methods selected for comparison were as follows. **FLoRA Wang et al. (2024b)**: A state-of-the-art federated fine-tuning architecture for heterogeneous environments. **FedIT Zhang et al. (2024a)**: A leading method for homogeneous environments based on FedAvg. **Centralized Hu et al. (2022)**: LoRA-based fine-tuning method. **Medalpaca Han et al. (2023)**: A conversational medical large model based on Llama. **AlpaCare Zhang et al. (2023)**: A instruction-finetuning medical large model based on Llama. We also included Llama-7B Touvron et al. (2023a) and TinyLlama Zhang et al. (2024b) as benchmark models.

## 4.2 GENERAL SCENARIOS PERFORMANCE

In general scenarios, we compared FF-LoRA with baseline methods across several environments. As shown in Table 1, FF-LoRA outperforms both FedIT and FLoRA in both homogeneous and heterogeneous settings. On the MMLU test set, FF-LoRA exhibited significant performance significant, surpassing FLoRA by 1.85% in a heterogeneous environment using Llama-7B. This demonstrates the efficacy of our method in question-answering tasks. Notably, in open-ended Wizard MT-Bench, our federated approach even surpassed the centralized LoRA method, suggesting that federated learning, through collaborative training across clients, enhances performance in open-ended tasks.

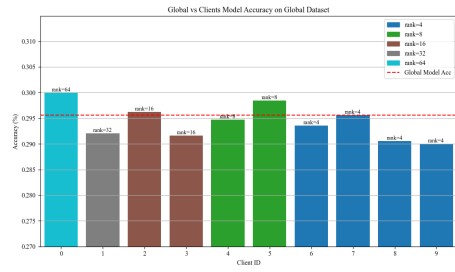

Figure 5: Comparison the performance of global model and client models based on Llama-7B.

Further, we also compared client models with the global model based on the Wizard dataset in a heterogeneous environment, as shown in Figure 5. The global model outperformed most client models across general datasets. When the rank was set to 64, the client model achieved a peak performance of 30.5, whereas the client with a rank of 16 had the lowest accuracy at 29.16. This indicates that the global model, by aggregating features

from multiple clients, provides more stable performance in general tasks compared to client models, which vary in performance depending on their focus on personalization.

### 4.3 FedBridge in Medical Scenarios

Based on the complex background of the medical domain introduced in Figure 1, to validate the effectiveness of our method in this scenario. We employed the FedBridge architecture, which integrating domain expert fine-tuned via PEFT with domain knowledge based on RAG, we leveraged the advantages of both to solve intricate problems in the healthcare domain, all while protecting user data privacy. We compared our approach with existing Llama and other optimized models, as well as federated methods. Evaluations were conducted at both the server and client levels.

#### 4.3.1 Server-Side Comparisons

We first evaluated the server-side global model using three publicly available closed-ended QA datasets: MedQA, MedMCQA, and PubMedQA. Additionally, the iCliniq dataset, which includes doctor-patient dialogues, was used to assess the model's ability to handle open-ended QA tasks.

Table 2: Comparison of FedBridge with baseline methods across various metrics and experimental settings.

| Policy | Method | Close-Ended Benchmark | | | | Open-Ended Benchmark | | |
|--------|--------|-------|---------|----------|------|----------|--------|------|
| | | MedQA | MedMCQA | PubmedQA | AVG | ChatGPT4 | Claude | AVG |
| Centralized | Llama-7B | 35.4 | 32.3 | 60.7 | 42.8 | 35.7 | 40.8 | 38.3 |
| | Alpacare | 34.5 | 31.3 | 59.3 | 41.7 | 49.7 | 53.4 | 51.6 |
| | Medalpaca | 36.7 | 30.2 | 63.2 | 43.4 | 48.8 | 55.7 | 52.3 |
| | PEFT&RAG | 38.5 | 33.5 | 68.6 | 46.9 | 59.4 | 60.9 | 60.2 |
| Fed | FLoRA | 33.8 | 30.6 | 57.7 | 40.7 | 48.3 | 53.2 | 50.8 |
| | FF-LoRA | 35.3 | 31.8 | 60.3 | 42.5 | 50.6 | 55.1 | 52.9 |
| | FedBridge | 38.1 | 33.7 | 66.5 | 46.1 | 58.7 | 62.8 | 60.8 |

The comparison results, presented in Table 2, demonstrate that our model, combining PEFT and RAG, performs best in both benchmark and open question-answering tasks when compared with other medical LLMs in a centralized environment. In a federated homogeneous environment, FedBridge outperforms both FLoRA and FF-LoRA models. Particularly in open-ended Q&A tasks, FedBridge even surpasses centralized models, illustrating that federated learning significantly enhances model generalization.

Interestingly, in the close-ended benchmark, Llama-7B outperformed the Alpacare model, which was specifically fine-tuned for a particular task. However, in generated Q&A tasks, the performance of general LLM LLaMa-7B often lags behind that of task-specific fine-tuned models. For example, in the open-ended benchmark dataset, the fine-tuned Alpacare model scored 51.55, while Llama-7B only scored 38.25. This discrepancy may be due to the fact that Q&A tasks are inherently closed-ended, with answers already provided among a limited set of options, whereas open-ended Q&A tasks require the model to predict the correct answer from a large pool of potential options, and sometimes even without any predefined options. This discrepancy highlights the advantage of domain-specific fine-tuning for open-ended tasks, underscoring the importance of training or fine-tuning domain-specific large models.

#### 4.3.2 Client-Side Comparisons

To address data heterogeneity, as depicted in Figure 1, we compared the global model with client models in both IID and Non-IID settings. In the IID experiments, where the Clinic dataset served as the global evaluation dataset, the global model exhibited greater stability than the client models, with the exception of some clients, as shown in Figure 6 left. This suggests that the global model offers superior stability in general tasks.

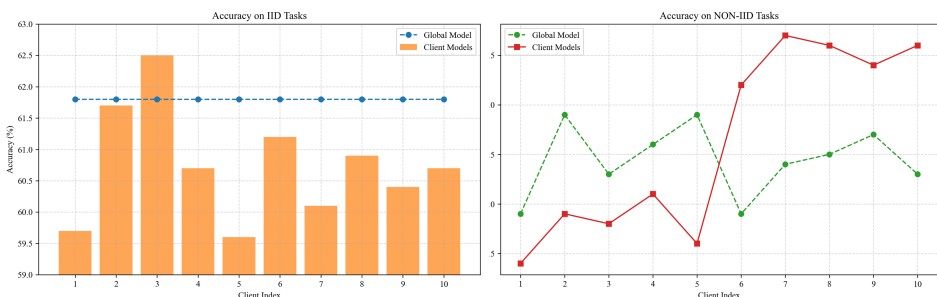

Figure 6: Performance comparison of global model and client models on IID and Non-IID.

In the Non-IID control experiments, we used each client's validation set as the evaluation dataset to form a Non-IID control group. The results, presented on the right of Figure 6 right, show that in general hospitals, the global model outperformed the client models. However, in specialized hospitals, client models significantly outperformed the global model. This finding validates the issue of client drift caused by data heterogeneity and further demonstrates the effectiveness of our method in addressing this challenge.

## 5 RELATED WORK

**FL & PEFT:** The integration of FL with PEFT has attracted considerable attention in recent years. Flat-LoRA Li et al. (2024) highlights that existing LoRA methods, constrained by low-rank matrix parameters, focus primarily on low-dimensional subspaces, which neglect the sharp directions in the parameter space and thus impair the model's generalization capability. FedIT Zhang et al. (2024a) introduces challenges posed by the noise during aggregation in the FedAVG McMahan et al. (2017) process. FLoRA Wang et al. (2024b) proposes effective solutions to the issues of heterogeneous rank and aggregation noise within the FL framework, but due to the rank heterogeneity, it requires frequent reinitialization of client models, leading to suboptimal performance Bian et al. (2024). FlexLoRA Bai et al. (2024) utilizes singular value decomposition (SVD) to manage resource and task heterogeneity across clients.

**FL & RAG**:Several studies have explored the integration of RAG into federated learning frameworks. Wang et al. (2024a) proposed the FeB4RAG method, which designs a federated query dataset specifically for retrieval-based question-answering tasks. This framework utilizes data from multiple clients to support efficient query processing. Mao et al. (2025) further proposed the FedE4RAG architecture to enable collaborative training of the embedding model across clients. These efforts highlight the potential of combining FL with RAG to preserve data privacy while enhancing retrieval accuracy and model performance.

## 6 CONCLUSION

In this paper, we introduce the FedBridge architecture, which synergistically integrates PEFT and RAG via federated learning to address the challenges posed by complex domain-specific tasks. Through the design of FF-LoRA, we fuse global and personalized features, thereby enhancing the model's ability to comprehend domain-specific knowledge while preserving personalized client features. The effectiveness of our proposed operators was validated in general scenarios. In the practical healthcare scenario, we developed an authoritative knowledge base and clients local case bases, which, when integrated via the federated architecture, effectively combine the advantages of fine-tuning and RAG. This bidirectional integration ensures consistency between the model's reasoning and generation components and the authoritative, domain-relevant knowledge. Experimental results demonstrate that FedBridge yields significant improvements in both close-ended domain and open-ended domain tasks, while also mitigating the client drift problem that is prevalent.

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

## A    Appendix

### A.1    The Communication Overhead of FLORA

As discussed in Section 4, the increase in the number of parameters has effectively mitigated the sharpness problem inherent in LoRA, which arises from insufficient parameterization. Simultaneously, our approach retains the personalization features of individual clients and integrates global features derived from federated learning. To assess the impact of this parameter increase on communication overhead, we compared FF-LoRA with full fine-tuning and FLoRA. The results are presented in Table 3.

Table 3: Comparison of communication and train overhead between FF-LoRA, FLoRA, and full fine-tuning methods.

| Method | Training | GPU size | Communicated |
|---|---|---|---|
| Full Finetuning | 6746M | 37386MiB | 6746M |
| FLoRA | 8.38M | 17324MiB | 92.27M |
| FFLoRA | 8.38M | 19280MiB | 92.27M |

Contrary to initial expectations, the results demonstrate that our method does not increase training or communication parameters compared to FLoRA. This can be attributed to the fact that, in the first stage, both FF-LoRA and FLoRA use the same training parameters. In the second stage, when the local and global parameters are integrated, the global parameters are frozen. Consequently, while our approach increases memory usage for the global weights, it does not introduce additional overhead for activation values or gradient computation, thus minimizing the computational burden. Regarding communication, our method only uploads the local weights of the client and downloads the global model weights, resulting in communication overhead equivalent to that of FLoRA.

### A.2    Environments, Datasets and Metric

**Computer Resources** For LoRA fine-tuning on all the models, we use 8 NVIDIA A100(40GB) GPUs.

#### A.2.1    Domain Datasets

**HealthCareMagic-100K**. A collection of 100,000 real patient–doctor dialogues scraped from HealthCareMagic.com, cleaned/anonymized and formatted for supervised fine-tuning. Compared with synthetic instruction sets, it captures colloquial symptom descriptions and physician reasoning, and was the primary training corpus for ChatDoctor. In practice it's often split into instruction–output QA pairs for tuning.

**iCliniq-10K**. An independent set of 10,000 real consultations from iCliniq.com. In ChatDoctor, it's used mainly as a held-out evaluation/validation source (and sometimes additional training), offering distribution shift relative to HealthCareMagic. We convert conversations to QA pairs for scoring.

**MedAlpaca**. A 160k-entry instruction-style corpus assembled for medical LLM fine-tuning. It blends: (1) rephrased Anki medical flashcards (33,955 QA); (2) StackExchange QA from five forums—Academia (39,633), Biology (7,482), Fitness (3,026), Health (1,428), Bioinformatics (906); and (3) WikiDoc QA—Living Textbook (67,704) and Patient Information (5,942). The authors fine-tune LLaMA models on this mix and report improvements on USMLE-style evaluations.

**Disease database**. This offline disease knowledge base, with entries covering fields such as "disease - symptoms - related tests/treatments - possible medications," can be continuously updated without requiring model retraining. The repository states that this library covers over 700 diseases and is used in conjunction with ChatGPT to synthesize approximately 5,000 doctor-patient conversations (GenMedGPT-5k) to enhance training.

**National Health Service**. This dataset consists of 24,665 command-answer pairs automatically generated by OpenGPT based on "Conditions A–Z" entries on the NHS UK website. The source content covers key points such as symptoms, causes, treatment, self-management, and when to seek medical attention. This dataset is used to construct a fine-tuned corpus of evidence-based

medical instructions. Data is sourced from the official NHS disease information pages to ensure standardization and consistency of medical descriptions.

**PubMed Abstract**. Clinical practice guidelines, systematic reviews/review papers, and publications from authoritative institutions (such as guidelines and monographs on the NCBI Bookshelf) together constitute the main carriers of "authoritative definitions/delimitations." Among them, clinical practice guidelines are defined by the IOM as recommendation statements based on systematic evidence evaluation for optimizing diagnosis and treatment decisions. They usually carry the definition of the disease, diagnostic criteria, and management pathways, and are the most core "authoritative" text form.

### A.2.2    GENERAL DATASETS

**Dolly**. The Dolly dataset is an open-source dataset with 15k text samples generated by Databricks employees. The topics include brainstorming, classification, closed QA, generation, information extraction, open QA, and summarization.

**Alpaca**. The Alpaca dataset contains 52K instruction-following data used for fine-tuning the Alpaca model. This dataset is believed to be diverse enough for fine-tuning LLMs.

**Wizard**. The Wizard dataset we use is the training data of the WizardLM model. it includes 70k pairsof instructions and outputs. The Wizard dataset generally featuresmore complex instructions compared to the other datasets. Its fine-tuning results are typically better, which has been confirmed by our experiments, especially those evaluated by the MT-bench scores.

**ShareGPT**. The ShareGPT dataset is a collection of approximately 52,000 conversations scraped via the ShareGPT API. The conversations in ShareGPT include both user prompts and responses from ChatGPT. In our experiments, we split the conversation dataset into question-answering pairs.

**MMLU test set**. The MMLU dataset is a widely used question-and-answer dataset in LLM fine-tuning. It has 14,024 questions in 57 different subjects, which can evaluate the logical reasoning capabilities of LLMs. We selected 1444 samples from the dataset for a quick and comprehensive evaluation.

**MT-bench**. MT-bench is a set of challenging multi-turn open-ended questions for evaluating chat assistants. It evaluates the performance of LLMs by using the GPT-4 API to score the LLM-generated conversations. LLMs that behave more like GPT-4 will receive higher scores.

### A.2.3    METRIC

In the medical Open-Ended Benchmark scenario, it is difficult to have a fixed evaluation method due to the diversity of generated answers. We followed established evaluation protocols and employed GPT-4 and Claude to evaluate the performance of different combinations on benchmark tests. For each given query, we constructed an evaluation list comprising the query question, reference answer, method_pred. GPT-4 was prompted to assign a score out of 100 to each of the four assistants based on the reference answer, accompanied by an explanation. To enhance evaluation accuracy, we performed multiple assessments by exchanging the order of model assistant responses using ChatGPT-4 and Claude and averaged the results.

### A.2.4    NON - INDEPENDENT IDENTICALLY DISTRIBUTION (NON-IID)

: Data heterogeneity is a significant challenge in FL framework, particularly when local data distributions deviate from global distributions. SloRA Babakniya et al. (2023) demonstrates that data heterogeneity exacerbates the performance gap between full fine-tuning and PEFT, with the negative impact on PEFT becoming more pronounced as heterogeneity increases. Furthermore, the full fine-tuning method used in the first stage of SloRA still imposes additional computational burdens on clients. To address the discrepancies between local data and global data, Qi et al. (2024) introduces a dual LoRA structure FDLoRA, which integrates global and local LoRA parameters in a certain proportion. Similarly, FedDPA Long et al. (2024); Yang et al. (2023) proposes federated dual personalized adapters that use an alternating learning approach to tackle both personalization needs and distribution shifts during testing. FedSA-LoRA Guo et al. (2024), based on the feature differences of matrices $A$ and $B$, updates matrix $A$ through federated aggregation while keeping matrix B updated locally, more effectively preserving client-specific information.

## A.3 Hyperparameter Details

In all our experiments, the learning rate for fine-tuning is set to 0.0003; the batch size is 128 and the micro batch size is 16. Given the substantial size of both the dataset and the models (e.g., LLaMA-7B), federated fine-tuning requires significant computational resources and time. To balance computational feasibility and experimental observation, we chose a limited number of communication rounds (typically 3, and sometimes as few as 1) and set the number of local training epochs to 1 per round. This design allows us to observe the effects of federated optimization while avoiding excessive overfitting, which is a known issue when using the MMLU dataset with large models and datasets. Fewer training rounds help ensure that the observed phenomena are meaningful and not dominated by overfitting.

## A.4 Supplementary Experiment Results

In addition, we also used a series of Llama foundation models (including Llama1, Llama2 and Llama3) based on the Wizard dataset to evaluate existing methods. Table 1 show that federated paradigms consistently outperform centralized LoRA; notably, FFLoRA delivers stable additional gains in both homogeneous and heterogeneous settings. During the Llama2 stage, improvements under heterogeneity are confirmed; the homogeneous setting does not uniformly degrade—FedIT decreases slightly whereas FFLoRA increases—suggesting that the effect is driven by method–backbone compatibility and hyperparameter sensitivity rather than overfitting induced by higher model complexity. In the Llama3 stage, performance becomes more stable and improves overall; however, marginal gains under homogeneity begin to saturate (FFLoRA +0.68), while improvements under heterogeneity remain substantial (FFLoRA +2.67). This underscores both the intrinsic difficulty of heterogeneous distributions and the necessity and promise of federated learning in such scenarios.

In summary, stronger backbones do not automatically resolve distributional shift: gains on homogeneous data tend to saturate, whereas federated personalization and knowledge fusion—particularly FFLoRA—continue to yield sizable improvements under heterogeneity.

| Model | Policy | Method | Wizard |
|---|---|---|---|
| LLama | Centralized | LoRA | 31.55 |
| | Home | FedIT | 33.47 |
| | | FFLoRA | 35.1 |
| | Heter | FLoRA | 27.7 |
| | | FFLoRA | 29.55 |
| Llama2 | Centralized | LoRA | 30.61 |
| | Home | FedIT | 33.13 |
| | | FFLoRA | 35.42 |
| | Heter | FLoRA | 29.69 |
| | | FFLoRA | 31.09 |
| Llama3 | Centralized | LoRA | 33.27 |
| | Home | FedIT | 35.21 |
| | | FFLoRA | 36.1 |
| | Heter | FLoRA | 31.57 |
| | | FFLoRA | 33.76 |

Table 4: Compare FF-LoRA with baselines in Llama foundation models

## A.5 LLMs in paper writing

We used LLMs only to perform grammar check and minor stylistic edits.

## A.6 LIMITATION

While our proposed ABCD matrix extension in FF-LoRA offers enhanced expressiveness and improved adaptation capabilities compared to the standard LoRA method, it is important to acknowledge several limitations. First, the introduction of two additional matrices (C and D) beyond the conventional A and B matrices does not increase the number of trainable parameters or communication overhead during federated training, as these matrices are designed to be efficiently computed and aggregated. However, this extension does incur additional memory overhead during both training and inference phases.

Specifically, the ABCD parameterization requires storing four matrices instead of two. This increased memory consumption can be particularly significant when dealing with large models or when multiple clients are training simultaneously in a federated setting. The additional memory overhead may limit the maximum batch size or the number of concurrent training processes that can be supported on hardware with constrained memory resources.

Furthermore, while the computational complexity of the ABCD extension remains comparable to the standard LoRA approach, the increased memory requirements may necessitate more sophisticated memory management strategies, especially in resource-constrained environments. This limitation should be carefully considered when deploying our method in production systems or when working with limited computational resources.

