# OpenReview forum: "FedBridge: Bridging Domain Experts and Domain Knowledge via a Federated Learning Framework for Controlled Model Personalization"
_ICLR.cc/2026/Conference — ICLR 2026 Conference Withdrawn Submission_

### Official Review · Reviewer_1xwR · 2025-10-30

**Soundness:** 2
**Presentation:** 3
**Contribution:** 3
**Rating:** 4
**Confidence:** 4

**Summary:**

The paper introduces FedBridge, a FL framework that integrates PEFT and RAG to address domain-specific challenges under privacy constraints. The key component, FF-LoRA, fuses server-level global knowledge with client-specific parameters to mitigate client drift in heterogeneous settings.

**Strengths:**

1. The paper addresses an important problem, how FL can effectively integrate multiple domain experts while respecting privacy constraints.
2. The framework is well-structured, with clear organization from motivation, methodology, to experiments, making the paper easy to follow.

**Weaknesses:**

1. Some figures (e.g., Figure 1, Figure 5, Figure 6) use font sizes that are too small.
2. The novelty is somewhat limited: the first part (federated training of LoRA) does not substantially differ from prior federated PEFT methods, while the second part (RAG) is largely orthogonal to FL and not clearly integrated as a core innovation.
3. The practical motivation is unclear. Why should a system combine decentralized LoRA training with a seemingly centralized RAG module? It is not evident whether such a deployment scenario exists in real-world.
3. In the experiments, only one sota baseline is compared under each setting. Either a broader comparison with multiple strong baselines or a justification for the current selection is needed.

**Questions:**

1. Give an example to illustrate the corresponding real-world applications of this method.
2. After the stage 2 training of the LORA, how is it used? Will the model index it from the client side?
3. It is suggested to incorporate the more SOTA methods.

---

### Official Review · Reviewer_ogn6 · 2025-10-30

**Soundness:** 2
**Presentation:** 1
**Contribution:** 1
**Rating:** 2
**Confidence:** 4

**Summary:**

The paper proposes FedBridge, which aims to integrate federated learning (FL), parameter-efficient fine-tuning (PEFT) via a variant called FF-LoRA, and retrieval-augmented generation (RAG). This paper shows that FedBridge alleviates client drift in FL, mitigates hallucination in RAG, and enhances domain-specific personalization, with experiments conducted on general and medical datasets.

**Strengths:**

This work explores a unified federated approach to jointly mitigate personalization and hallucination problems, yielding preliminary yet promising results toward integrating model adaptation with retrieval across domains.

**Weaknesses:**

1. The motivation of this paper is not clear. The paper attempts to tackle two problems—client drift in federated fine-tuning and hallucination in RAG systems—but fails to establish a coherent motivation for combining them. The rationale for why merging PEFT and RAG within a single federated framework is necessary or even meaningful is poorly articulated. The paper reads like two half-developed ideas awkwardly glued together rather than a unified contribution.
2. The formulation of FF-LoRA and its fusion operation $\Delta W=F_{global} F_{personalized}$
 (Eq. 4) lacks any mathematical justification. Multiplying two matrices to represent “feature fusion” is arbitrary and unsupported by either theoretical analysis or empirical ablation. The description of global and local model aggregation is confusing, and the matrix update seems ad-hoc, with no insight into why it improves expressiveness or stability. The claims about mitigating “sharpness” are unverified.
3. The RAG part of the framework is equally problematic. Equations (5)-(7) are ambiguous:
    - It is unclear what the embedding function $E(.; \Theta)$ accepts as input, as "q" and "Latent" are of very different modalities, where one is query and the other one is the latent embedding.
    - The definition of $TopK(⋅)$ is missing.
    - The choice of using $e_L$ for $D_{auth}$ and $e_q$ for $D_{case}$ appears arbitrary.
Overall, the RAG design is described without any detail about retrieval index construction, negative sampling, or retriever-generator alignment training.
4. The experimental validation is not convincing. Although the authors compare against Flora and claim improvements after introducing RAG, they fail to compare against a standard PEFT + RAG baseline under heterogeneous client settings. Moreover, the paper overlooks the computational and communication overhead of FedBridge relative to other baselines. Given its additional modules, the proposed approach likely incurs significantly higher memory and time costs, which should be quantitatively analyzed

**Questions:**

Please check Weaknesses.

---

### Official Review · Reviewer_TVfR · 2025-10-31

**Soundness:** 2
**Presentation:** 2
**Contribution:** 2
**Rating:** 2
**Confidence:** 3

**Summary:**

This paper introduces FedBridge, a novel framework designed to address key challenges in applying Large Language Models (LLMs) to domain-specific applications, particularly in privacy-sensitive domains like healthcare. FedBridge synergistically integrates Federated Learning (FL), Parameter-Efficient Fine-Tuning (PEFT), and Retrieval-Augmented Generation (RAG). Extensive experiments in both general and medical domains demonstrate that FedBridge outperforms strong baselines in various settings (IID and Non-IID), validating its effectiveness and robustness.

**Strengths:**

1. The proposed FF-LoRA is a well-motivated solution to client drift, and the two-stage knowledge fusion strategy is sound.
2. The paper validates its method thoroughly across both general and medical domains, using multiple models, datasets, and data distributions.
3. The work directly addresses pressing issues in applying LLMs to sensitive, real-world domains like healthcare, making it highly applicable.

**Weaknesses:**

1. The definitions of some symbols used in the paper are somewhat ambiguous. For instance, the mathematical notation for the global model (Global_C, Global_D) is a weak point and needs clearer definition.

2. This paper lacks the comparison with existing RAG baselines. It is better to compare the proposed methods with recent FL+RAG methods (e.g., FedE4RAG, mentioned in the related work).

3. The paper shows that client models can outperform the global model in specialized hospitals but does not deeply analyze the characteristics of these "strong" client models or the trade-off between personalization and global generalization.

4. The motivation for combining PEFT and RAG is not well explained. The current method is more like a direct combination of the two methods.

5. The font size of the captions for the pictures in the paper is too small.

**Questions:**

1. Can the authors provide some comparisions with current FL+RAG methods?

2. In Equation (2), what are the precise dimensions of Global_C and Global_D? How are they determined (e.g., as a function of the number of clients N, the client rank r, or a new hyperparameter c)?

3. What does $\bigoplus$ mean in equation (2)? How was the dimension transformed into the dimensions of Global_C and Global_D?

4. What is the total number of clients, $n$ or $N+1$? It seems that in Equation(2), the total number of clients equals $N+1$, but in lines 196-202, the total number of clients seems to be $n$.

5. In the Non-IID results (Figure 6, right), client models in specialized hospitals significantly outperform the global model. Does this suggest that FF-LoRA's personalization is sometimes too strong? How do the authors conceptualize the optimal balance between global knowledge and local specialization in this framework?

6. The retriever's embedding model E(·;Θ) is mentioned to be fine-tuned locally and aggregated globally. Could the authors elaborate on the training details? Were there any challenges in achieving a consistent embedding space across clients with heterogeneous case bases?

7. Appendix A.6 mentions the ABCD extension increases memory usage. For a typical client (e.g., with a single A100 GPU), what is the practical impact on the maximum trainable model size or batch size? Are there strategies to mitigate this?

---

### Official Review · Reviewer_kXpW · 2025-10-31

**Soundness:** 2
**Presentation:** 2
**Contribution:** 2
**Rating:** 4
**Confidence:** 3

**Summary:**

The paper proposes FedBridge, a federated framework that combines a two-stage LoRA-style fusion operator called FF-LoRA with a retrieval-augmented generation (RAG) pipeline and dual knowledge bases. The design aims to reduce client drift under data heterogeneity and to improve domain fidelity. Empirical results report gains on general benchmarks and medical benchmarks.

**Strengths:**

1. The work targets a critical real-world challenge—applying LLMs to privacy-sensitive domains where data cannot be centralized. This aligns with growing demand for federated AI solutions in industry and academia.
2. Tackles a timely and practically important problem: combining FL, PEFT, and RAG for domain-sensitive LLMs.

**Weaknesses:**

1. The paper reads to an extent as "LoRA + RAG in FL" without a clear novel principle or theoretical insight explaining why this combination should succeed beyond engineering intuition.
2. The evaluation does not compare against a federated baseline that uses standard LoRA + the same RAG retrieval pipeline (i.e., FL+RAG without the FF fusion). Without this, it's impossible to know if gains are from FF-LoRA, the RAG design, or dataset/metric differences.
3. No systematic ablation isolating: (a) RAG alone (no FF-LoRA; standard LoRA or full-tune), (b) the dual-knowledge base vs single KB, (c) the latent-feature guided retrieval vs standard retrieval.
4. Missing or under-specified items that are essential for reproducibility and fair comparison: number of clients, client selection ratio, LoRA rank values for FF-LoRA and baselines

**Questions:**

1. Please see the weaknesses above.
2. How robust is the retrieval to noisy/vague user queries? It would be helpful to provide a small robustness study (noisy paraphrases, shorter queries).

---

### Note · Authors · 2026-01-04

I have read and agree with the venue's withdrawal policy on behalf of myself and my co-authors.